# Broad-complex transcription factor mediates opposing hormonal regulation of two phylogenetically distant arginine kinase genes in *Tribolium castaneum*

Nan Zhang[1], Heng Jiang[1], Xiangkun Meng[1], Kun Qian[1], Yaping Liu[1], Qisheng Song [2], David Stanley[3], Jincai Wu[1], Yoonseong Park[4] & Jianjun Wang[1✉]

The phosphoarginine-arginine kinase shuttle system plays a critical role in maintaining insect cellular energy homeostasis. Insect molting and metamorphosis are coordinated by fluctuations of the ecdysteroid and juvenile hormone. However, the hormonal regulation of insect arginine kinases remain largely elusive. In this report, we comparatively characterized two arginine kinase genes, *TcAK1* and *TcAK2*, in *Tribolium castaneum*. Functional analysis using RNAi showed that *TcAK1* and *TcAK2* play similar roles in adult fertility and stress response. TcAK1 was detected in cytoplasm including mitochondria, whereas TcAK2 was detected in cytoplasm excluding mitochondria. Interestingly, *TcAK1* expression was negatively regulated by 20-hydroxyecdysone and positively by juvenile hormone, whereas *TcAK2* was regulated by the opposite pattern. RNAi, dual-luciferase reporter assays and electrophoretic mobility shift assay further revealed that the opposite hormonal regulation of *TcAK1* and *TcAK2* was mediated by transcription factor Broad-Complex. Finally, relatively stable AK activities were observed during larval-pupal metamorphosis, which was generally consistent with the constant ATP levels. These results provide new insights into the mechanisms underlying the ATP homeostasis in insects by revealing opposite hormonal regulation of two phylogenetically distant arginine kinase genes.

[1] College of Horticulture and Plant Protection, Yangzhou University, 225009 Yangzhou, China. [2] Division of Plant Sciences, University of Missouri, Columbia, MO, USA. [3] USDA/Agricultural Research Service, Biological Control of Insects Research Laboratory, Columbia, MO 65203, USA. [4] Department of Entomology, Kansas State University, Manhattan, KS, USA. ✉email: wangjj@yzu.edu.cn

Phosphagen kinases (PKs) comprise a large family of proteins that catalyze the reversible transfer of the γ-phosphoryl group of ATP to guanidine substrate, yielding ADP and a phosphorylated guanidine commonly known as a phosphagen. Eight phosphagens and corresponding PKs are recognized in animals, including kinases for creatine (CK), arginine (AK), hypotaurocyamine, glycocyamine, thalessemine, opheline, lombricine, and taurocyamine[1–3]. Phylogenetic analyses revealed two distinct clusters of PKs, a CK supercluster and an AK supercluster[4,5]. AK enzymes from the echinoderm *Stichopus japonicus* and annelid *Sabellastarte indica* are apparently homologous to those of vertebrate CK and they clustered into the CK group, indicating that AK evolved at least twice during the evolution of guanidino kinases[2,5].

Although CK is the only PK in vertebrates, invertebrates express various PKs, including the most widely distributed AK. AK has been identified in bacteria, protozoans including ciliates, trypanosoma, and choanoflagellates, and a range of invertebrates such as arthropods, molluscs, nematodes, cnidarians, and poriferans[3,6–11]. The presence of AK in these invertebrates indicates its ancient origin. A single AK gene was identified in insects, but later bioinformatics analysis revealed two AK genes in the genomes of *Anopheles gambiae*, *Aedesae gypti*, and *Apis mellifera*. Generally, insect AKs can be separated into two groups[3,12]. Group 1 consists of typical AKs from various insect species. Group 2 includes AK2s in *A. gambiae*, *A. aegypti*, *A. mellifera*, and *Cissites cephalotes*. Their functional significance is still unknown.

ATP drives virtually all energy-dependent activities in organisms[13]. Regulation of intracellular ATP concentration is a homeostatic process in which cells maintain a constant level of ATP, even when the demand for or supply of ATP fluctuates[14]. PKs act in maintaining cellular energy homeostasis as ATP-buffering systems in periods of high-energy demand or energy supply fluctuations[1]. Aberrant PK levels may impair cell viability under normal or stressed conditions and induce cell death[15–18]. However, the molecular mechanisms underlying the regulation of PKs have not been investigated sufficiently both in vertebrates and in invertebrates.

Here, we investigate the functions of two ATP homeostasis-related genes, *TcAK1* and *TcAK2*, and their hormonal regulatory mechanisms in the red flour beetle, *Tribolium castaneum*. We report molecular evidence that *TcAK1* and *TcAK2* play similar roles in highly energy-demanding processes, such as stress response and adult fertility, and the opposing regulation of *TcAK1* and *TcAK2* by 20-hydroxyecdysone (20E) and juvenile hormone (JH) is mediated by insect-specific transcription factor Broad-Complex (BR-C). Our results indicate that *T. castaneum* might maintain ATP homeostasis by opposing hormonal regulation of two phylogenetically distant AK genes.

## Results

### cDNA cloning and sequence analysis of *TcAK1* and *TcAK2*.
Full-length cDNA sequences of *TcAK1* and *TcAK2* were amplified. The *TcAK1* cDNA has 1471 bp with a 52-bp 5′-untranslated region (UTR) and a 1068 bp of ORF encoding a 355-amino acid protein with a molecular mass of 39.9 kDa, an isoelectric point of 5.92, and a 351-bp 3′-UTR (Supplementary Fig. 1a). The *TcAK2* cDNA has 1283 bp with a 56-bp 5′-UTR, a 1116 bp of ORF encoding a 371-amino acid protein with a molecular mass of 41.9 kDa, an isoelectric point of 8.39, and a 111-bp 3′-UTR (Supplementary Fig. 1b). A putative polyadenylation signal (AATAAA) was found in 13 bp upstream of a poly (A) tail in *TcAK1*, but not in *TcAK2*. The phylogenetic tree shows that TcAK1 clusters within group 1 and TcAK2 falls into group 2 AKs

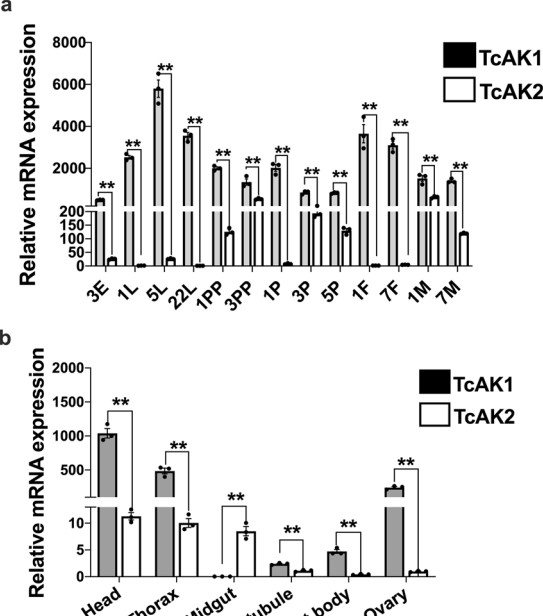

**Fig. 1 mRNA expression levels of *TcAK1* and *TcAK2* in different development stages and different tissues of 7-day-old female adults.** **a** Relative mRNA expression levels of *TcAK1* and *TcAK2* in different development stages. **b** Relative mRNA expression levels of *TcAK1* and *TcAK2* in different tissues of 7-day-old female adults. Data are expressed as mean ± SEM ($n = 3$ biologically independent replicates). Asterisks indicate differences statistically significant at **$P < 0.01$ (Student's *t*-test).

(Supplementary Fig. 2). Amino acid sequence alignment shows that TcAK1 shares 85.9% identity with AmAK1 in *A. mellifera*, 86.5% with BmAK in *Bombyx mori*, and 70.2% identity with TcAK2. TcAK2 shares 56.2% and 65.3% identities with CcAK in *C. cephalotes* and AmAK2 in *A. mellifera*, respectively (Supplementary Fig. 3). TcAK1 amino acids 270–276 and TcAK2 amino acids 286–292 feature the signature sequence patterns of ATP-guanidino kinases, CP(S/T)N(I/L)GT[9,19,20]. Key residues involved in the binding of ATP and the guanidine substrate are highly conserved (Supplementary Fig. 3).

### Genomic organization of *TcAK1* and *TcAK2*.
The genomic structures of *TcAK1* and *TcAK2* were predicted by comparing the full-length cDNA sequences with the genomic sequences retrieved from contigs in the whole-genome shotgun release for *T. castaneum* (Supplementary Fig. 4). A large 1247 bp intron occurs in the 5′-UTR 10 bp upstream from the ATG initiation codon in *TcAK1*. A shorter intron occurs within the coding regions of both genes at codons specifying amino acids 56 and 291, respectively. The 5′ donor and 3′ acceptor site sequences agree with the GT/AG consensus sequence[21].

### mRNA expression patterns of *TcAK1* and *TcAK2*.
Generally, the relative expression of *TcAK1* is substantially higher than *TcAK2* in all examined developmental stages and tissues of 7-day-old female adults except in the midgut (Fig. 1). The relative accumulation of mRNA encoding *TcAK1* was the highest in 5-day-old larvae and lowest in 3-day-old eggs (Fig. 1a). The highest accumulation of mRNA encoding *TcAK2* occurred in 1-day-old males, with lower accumulations in the other stages (Fig. 1a). Notably, compared to 22-day-old larvae (22L), *TcAK1* mRNA levels were significantly decreased by 1.42- and 2.31-fold in

**Table 1 Comparison of the kinetic constants among insect and mollusc arginine kinases.**

| Sources | Reference | $k_{cat}^{Arg}$ (s$^{-1}$) | $K_m^{Arg}$ (mM) | $k_{cat}/K_m^{Arg}$ (s$^{-1}$ mM$^{-1}$) | $k_{cat}^{p\text{-}Arg}$ (s$^{-1}$) | $K_m^{p\text{-}Arg}$ (mM) | $k_{cat}/K_m^{Arg}$ (s$^{-1}$ mM$^{-1}$) |
|---|---|---|---|---|---|---|---|
| *Tribolium castaneum* TcAK1 | This work | 172.7 ± 5.8 | 1.5 ± 0.02 | 112.9 | 22.00 ± 1.5 | 14.0 ± 0.05 | 1.5714 |
| *Tribolium castaneum* TcAK2 | This work | 12.7 ± 0.95 | 3.7 ± 0.04 | 3.5 | 146.1 ± 6.3 | 1.40 ± 0.10 | 104.357 |
| *Ctenocephalides felis* CfAK1 | [52] | ND | 1.27 ± 0.032 | ND | ND | 2.31 ± 0.05 | ND |
| *Ctenocephalides felis* CfAK2 | [52] | ND | 0.21 ± 0.026 | ND | ND | 0.74 ± 0.042 | ND |
| *Locusta migratoria manilensis* | [22] | 159.4 ± 6.2 | 0.951 ± 0.08 | 169 | ND | ND | ND |
| *Cissites cephalotes* | [12] | 2.02 ± 0.05 | 1.01 ± 0.07 | 2.01 | ND | ND | ND |
| *Crassostrea gigas* | [23] | 47.5 ± 2.05 | 0.35 ± 0.11 | 136 | ND | ND | ND |
| *Scapharca broughtonii* | | 72.1 ± 7.45 | 1.44 ± 0.28 | 50.07 | ND | ND | ND |

*Arg* arginine, *p-Arg* phosphoarginine-arginine, *ND* not determined.

1-day-old prepupa (1PP) and 3-day-old prepupa (3PP), respectively, while *TcAK2* mRNA levels were sharply increased by 99- and 593-fold in 1PP and 3PP, respectively, suggesting to us that *TcAK1* and *TcAK2* might be oppositely regulated by ecdysone during larval–pupal metamorphosis. Anatomical regulation of *TcAK1* and *TcAK2* expression was also observed with the highest expression level of both genes in head followed by thorax. The lowest expression level of *TcAK1* and *TcAK2* appeared at the midgut and fat body, respectively (Fig. 1b).

**Heterologous expression of TcAK1 and TcAK2 and characterization of the enriched recombinant enzymes**. We expressed TcAK1 and TcAK2 in bacteria and enriched the recombinant proteins (r proteins) using Ni$^{2+}$-NTA agarose chromatography. The proteins appeared as a single band with the expected molecular masses around 40 kDa (Supplementary Fig. 5). Western blots with an anti-His-tag antibody showed single protein band around 40 kDa (Supplementary Fig. 5).

Enzyme stereospecificity for chiral arginine substrates showed that TcAK1 and TcAK2 have marked preferences for L-arginine over D-arginine. Among the other potential guanidine substrates investigated, rTcAK1 showed an appreciable reaction rate of approximately 15% compared to L-arginine for L-Nα-acetyl-arginine and L-canavanine. rTcAK2 showed no significant activity for the other substrates tested (Supplementary Fig. 6).

The apparent $K_m$ constants for two AK substrates were about 1.5 mM [L-arginine ($K_m^{Arg}$)] and 14 mM [phospho-L-arginine ($K_m^{P\text{-}Arg}$)] for TcAK1 and about 3.7 mM ($K_m^{Arg}$) and 1.4 mM ($K_m^{P\text{-}Arg}$) for TcAK2. The corresponding $k_{cat}$ values for TcAK1 were 172.7 s$^{-1}$ for L-arginine and 22.0 s$^{-1}$ for phospho-L-arginine. The values for TcAK2 were 12.7 s$^{-1}$ for L-arginine and 146.1 s$^{-1}$ for phospho-L-arginine. The catalytic efficiency as $k_{cat}/K_m^{Arg}$ of TcAK1 is about 32.6-fold higher compared to that of TcAK2 (Table 1). The $k_{cat}/K_m^{Arg}$ of TcAK1 is similar to the values for the *Locusta migratoria manilensis* AK[22] and *Crassostrea gigas* AK[23]. The $k_{cat}/K_m^{Arg}$ values of TcAK2 was comparable to group 2 AK from *C. cephalotes*[12] (Table 1).

**Role of TcAK1 and TcAK2 in development**. Injection of dsRNA led to reduction of mRNA levels by 85.0 ± 10% for *TcAK1* and 80.6 ± 3% for *TcAK2* (Supplementary Fig. 7a). Larvae injected with dsTcAK2 showed no abnormalities and developed to normal adults. dsTcAK1 treatment led to externally visible developmental abnormalities in pupae and adults (Fig. 2). While 85.8 ± 4.6% of dsTcAK1-treated larvae could pupate (Fig. 3a), a number of late stage pupae showed wing deformities, failed to eclose and died (Fig. 2a), and 22.9 ± 3.9% of the pupae succeeded in eclosing (Fig. 3b), albeit with wing and cuticle deformation and early death (Fig. 2b).

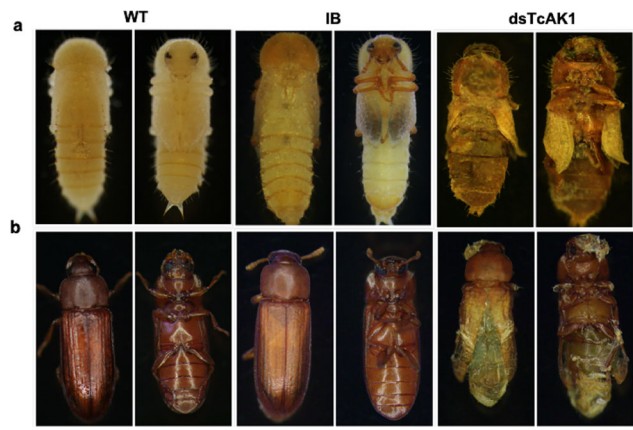

**Fig. 2 RNAi phenotypes of TcAK1. a** Injection of dsRNA for *TcAK1* resulted in defective larval–pupal metamorphosis. **b** Injection of dsRNA for *TcAK1* resulted in abnormal folding of the adult wings.

The effects of *TcAK1* and *TcAK2* on fecundity and fertility were further investigated after 2-day-old pupal RNAi treatments. The accumulations of mRNA encoding *TcAK1* and *TcAK2* were reduced by 83.4 ± 4.8% and 85.6 ± 3.6%, respectively, in 4-day-old females (Supplementary Fig. 7b). Compared with the controls (5.9 ± 0.44 eggs/day/pair for WT and 5.6 ± 0.48 eggs/day/pair for IB), silencing *TcAK1*, but not *TcAK2*, led to significantly reduced fecundity (3.7 ± 0.26 eggs/day/pair) (Fig. 3c). Injecting dsTcAK1 and dsTcAK2 reduced F1 egg hatching rate to 17.8 ± 2.94% (dsTcAK1) and 23.3 ± 1.93% (dsTcAK2) (Fig. 3d). It is broadly expected that most, certainly not all, healthy eggs hatch in a timely manner. Because separate injections of dsTcAK1 and dsTcAK2 lead to reduced hatching, we infer that both enzymes independently act in embryonic development.

We considered the possibility of sex-specific effects of the dsRNA treatments by performing reciprocal crosses. Pairs of dsTcAK1-treated females and control males produced about 4.3 ± 0.16 eggs per day, significantly less than controls (Fig. 3c). Similarly, crossing dsTcAK1-treated males with control females led to reduced egg laying, down to 4.0 ± 0.28 eggs per day (Fig. 3c). Our results also showed that the influence of silencing *TcAK1* and *TcAK2* on embryonic development was not sex-specific (Fig. 3d).

**Stress-induced mRNA and protein expression of TcAK1 and TcAK2**. We determined the influence of applied stressors on transcript levels of mRNAs encoding *TcAK1* and *TcAK2*. Compared with controls, accumulations of mRNAs encoding *TcAK1* and *TcAK2* were not significantly altered during the first 6 days of fasting. *TcAK1* expression declined on days 12–14. *TcAK2*

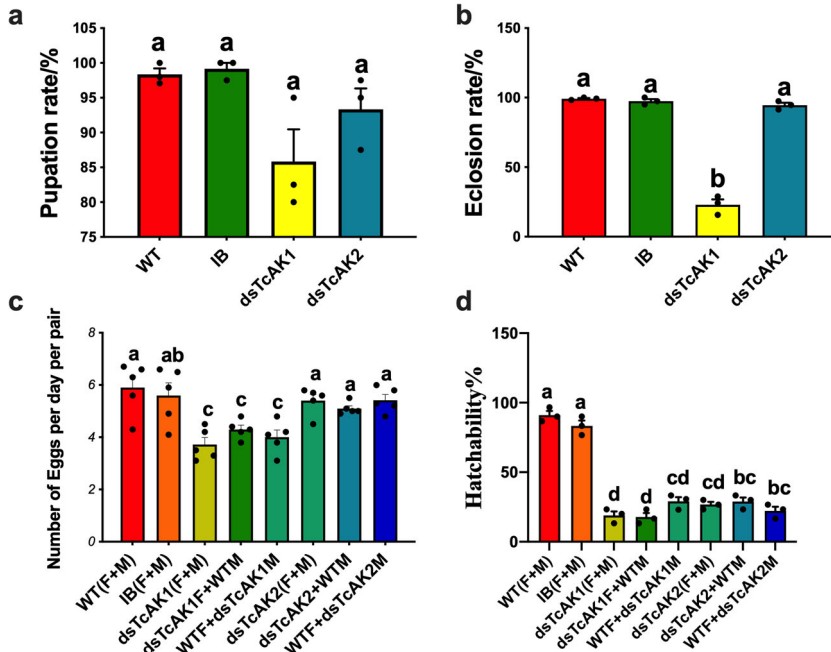

**Fig. 3 The role of *TcAK1* and *TcAK2* in development.** The effects of knockdown of expression of *TcAK1* and *TcAK2* on the rates of **a** pupation, **b** eclosion, **c** fecundity, and **d** fertility. Data are expressed as mean ± SEM ($n = 3$ or 5 biologically independent replicates). Histogram bars annotated with the same lowercase letters are not significantly different (one-way ANOVA, $P < 0.05$).

expression declined after fasting through days 8 to 14 (Supplementary Fig. 8a). Western blotting analysis results confirmed that starvation increases the protein abundance of TcAK1 from 2 to 4 days post-treatment (PT), subsequently, the expression level decreased significantly (Supplementary Figs. 8a and 9a). For the expression of TcAK2 protein, western blotting indicated that it was up-regulated after 4 days of treatment, and the subsequent treatment led to its significant down-regulation (Supplementary Figs. 8a and 9a).

The heat and cold stress treatment generally increased the expression levels of both genes. mRNAs encoding *TcAK1* and *TcAK2* increased by >665-fold (*TcAK1*) and >150-fold (*TcAK2*) after 1 and 2 h at 42 °C (Supplementary Fig. 8b). Expression of these two genes declined to nearly control levels at 12 h PT. Increased protein abundance was also observed after 2 h PT for TcAK1 and 4 h PT for TcAK2, respectively (Supplementary Figs. 8b and 9b).

Accumulations of mRNA encoding *TcAK1* increased by two-fold or less over the first 4 h of cold stress, then reached a high of about three-fold after 12 h. The accumulations of mRNA encoding *TcAK2* increased by about ninefold at 1 h and by about six-fold at 2 h in the cold, reached a peak of about 35-fold at 4 h, and declined to about 20-fold more than control levels (Supplementary Fig. 8b). Increased protein levels appeared after 4 h PT for TcAK1 as well as after 2 h PT for TcAK2, respectively (Supplementary Figs. 8b and 9c).

Paraquat treatment can lead to oxidative stress. We did not directly determine the influence of our paraquat treatments on oxidation products in beetle tissues, such as malondialdehyde (MDA) produced by the lipid peroxidation, and we ran this experiment based on literature reporting that exposure to paraquat leads to increased MDA. The results showed that paraquat treatments led to significantly increased accumulations of mRNAs encoding *TcAK1* and *TcAK2* (Supplementary Fig. 8c). In agreement with increased mRNA expression level, paraquat treatment resulted in the increase of protein abundance after 1 h

PT for TcAK1 and after 4 h PT for TcAK2, respectively (Supplementary Figs. 8c and 9d).

**TcAK1 and TcAK2 act in stress tolerance.** Adults treated with dsTcAK1 or dsTcAK2 suffered reduced survival rate following stress treatments. Compared to controls, survival was down to approximately 39% in the dsTcAK1-treated group and 53% in the dsTcAK2-treated group following heat stress (Supplementary Fig. 10a) and down to about 3% in the dsTcAK1-treated group or 25% in the dsTcAK2-treated group after paraquat treatments (Supplementary Fig. 10b). Suppressing *TcAK1* and *TcAK2* expression had no significant effect on the tolerance to starvation or cold stress at 4 °C.

**Subcellular localization of TcAK1 and TcAK2.** The finding of similar TcAK1 and TcAK2 functions in fertility and stress response raises the question of whether TcAK1 and TcAK2 have a similar subcellular localization pattern. We used immuno-fluorescence staining to characterize the subcellular localization of TcAK1 and TcAK2 in the midgut of female adults. We recorded a TcAK1 pattern of small dots in cytoplasm including mitochondria (Fig. 4a). The TcAK2 pattern was otherwise, with most staining in the cytoplasm near the cell edges, excluding mitochondria (Fig. 4a). Neither TcAK1 nor TcAK2 appeared in nuclei. Similar results were also obtained in immunofluorescence staining of other tissues including ovary and Malpighian tubule. Western blot of fractionated proteins revealed the same subcellular localization pattern of both proteins in females (Fig. 4b).

**Influence of ecdysteroid and JH on *TcAK1* and *TcAK2*.** The finding that *TcAK1* and *TcAK2* showed opposite expression pattern during molting and pupation led us to determine the role of ecdysteroid and JH in the regulation of expression of *TcAK1* and *TcAK2*. Separate injection treatments with dsTcphantom, dsTcshade, dsTcJHAMT, dsTcMet, and dsTcKr-h1 influenced

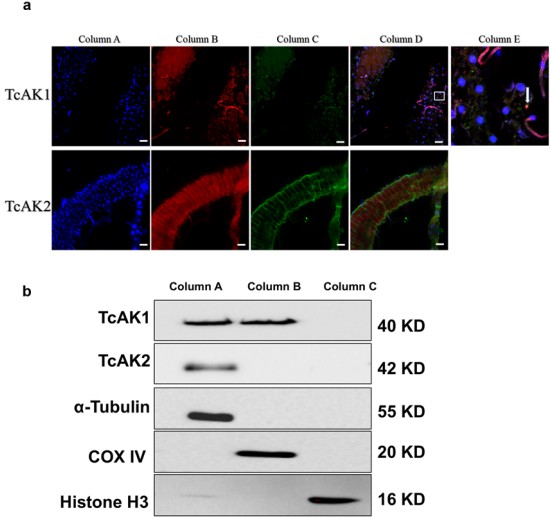

**Fig. 4 TcAK1 and TcAK2 have different subcellular localization patterns.**
**a** Immunofluorescence of TcAK1 and TcAK2. Midguts were attached to multi-well slides, fixed, permeabilized, and stained with DAPI to visualize the nuclei (column A), mitochondria-specific markers MitoTracker Red CMXRos (column B), and antibodies against TcAK1 or TcAK2 (column C). Merged and enlarged images are shown in column D and column E, respectively. **b** Western blots of fractionated cellular proteins, including cytoplasmic protein without mitochondria (Column A), mitochondrial protein (Column B), and nuclear protein (Column C). Western blot was conducted using antibody against TcAK1 or TcAK2 and subcellular-specific antibodies, including anti-α-Tubulin antibody (cytoplasmic marker), anti-COX IV antibody (mitochondrial marker), and anti-Histone H3 antibody (nuclear marker). Experiments were repeated at least three times with similar results.

relative accumulations of mRNAs encoding TcAK1 and TcAK2. mRNAs encoding TcAK1 were increased following dsTcphantom and dsTcshade injections and substantially decreased following dsTcJHAMT, dsTcMet, and dsTcKr-h1 treatments at day 6 PT. However, opposite TcAK2 expression patterns followed the dsRNA treatments (Fig. 5a).

In vivo and in vitro treatments with JH III and 20E were conducted to confirm the RNAi results. For 20E larval treatments, accumulations of mRNAs encoding TcAK1 were significantly down-regulated by at least 20% at the indicated times PT. mRNAs encoding TcAK2 were up-regulated from 30 min to 4 h, but not at 12 h PT (Fig. 5b). JH III treatment led to considerable increases in accumulations of mRNAs encoding TcAK1 up to the first 2 h PT. mRNAs encoding TcAK2 were significantly decreased over the 12-h experiment period (Fig. 5b). Our findings were validated at the protein level by western blot analysis (Fig. 5b and Supplementary Fig. 9e, f). The effects of JH III and 20E on the expressions of TcAK1 and TcAK2 were further verified by dual-luciferase assay in S2 cells, which showed that 20E treatment up-regulated TcAK1 promoter activity and down-regulated TcAK2 promoter activity, and JH III, on the other hand, showed the opposite effects (Fig. 5c).

The insect-specific transcription factor, Broad-Complex (BR-C), regulates a variety of developmental processes including larval-pupal metamorphosis. We found that TcBR-C expression is induced by 20E, and this induction is repressed by JH III in T. castaneum (Fig. 6a), we therefore hypothesized that BR-C might act in the opposite regulation of TcAK1 and TcAK2 by 20E and JH III. Consistent with this hypothesis, two prediction programs, JASPAR (http://jaspar.genereg.net/) and TFBIND (http://tfbind.hgc.jp/), predicted several putative BR-C-binding motifs in

promoter regions of TcAK1 and TcAK2 (Supplementary Table S2). RNAi was conducted to clarify the transcriptional regulation of TcAK1 and TcAK2 by BR-C. The injection of dsTcBR-C into 20L up-regulated TcAK1 expression whereas down-regulated TcAK2 expression at both mRNA and protein levels (Fig. 6b). Knockdown of TcBR-C expression also abolished the effects of 20E and JH III on both TcAKs expression (Fig. 6b). We further carried out a reporter assay using plasmids containing 5′-flanking regions of TcAK1 (from −2170 to +130 bp) and TcAK2 (from −2079 to +88 bp) in S2 cells. Dual-luciferase assay revealed that compared with the empty control without TcBR-C overexpression, TcBR-C-z1, TcBR-C-z2, TcBR-C-z3, TcBR-C-z4, and TcBR-C-z5 inhibited TcAK1 promoter-driven luciferase expression by 60%, 89.80%, 21.26%, 61.54%, and 72.22%, respectively (Fig. 6c), while enhanced TcAK2 promoter-driven luciferase expression by 1.40-, 12.24-, 2.19-, 1.83-, and 4.13-fold, respectively (Fig. 6c). We also transfected S2 cells with pAC-pGL-TcAK1/pAC-pGL-TcAK2 and pAC5.1-TcBr-C-z2 followed by incubation with 20E and JH III, which showed consistency with in vivo experiments (Fig. 6d). Electrophoretic mobility shift assay (EMSA) experiments further verified that TcBR-C-z2 directly binds to the TcAK1 and TcAK2 promoter regions to regulate their expression (Fig. 6e). Our interpretation is that the opposite hormonal regulation of TcAK1 and TcAK2 was primarily mediated by TcBR-C-z2.

To clarify whether BR-C is also involved in regulation of DmAK1 and DmAK2 genes in Drosophila, pAC-DmBR-C-zs were constructed and transfected into the S2 cell line. Overexpression of DmBR-C-Z3 down-regulated DmAK1 expression, whereas up-regulated DmAK2 expression (Fig. 7), which is consistent with our findings in T. castaneum. However, other BR-C isoforms had no effects on the mRNA expression of DmAKs.

**In vivo AK activities and ATP levels during larval–pupal metamorphosis and after hormonal treatment.** PKs regulate ATP homeostasis in subcellular compartments by transferring phosphates between phosphagen and adenine nucleotides. The opposite hormonal regulation of TcAK1 and TcAK2 might contribute to temporal ATP homeostasis, especially during larval–pupal metamorphosis. To test this idea, we first used exogenous hormone treatments to determine the effects of 20E and JH III on the ATP content of beetles. The ATP content was significantly decreased after treatment with 20E for 30 min and 1 h but recovered to normal levels at 4 h after treatment. JH III treatment increased the ATP content at 30 min and 1 h after treatment but showed no effects on the ATP levels later (Supplementary Fig. 11). These results support the function of AKs as an ATP buffering system during the molting phase. In vivo AK activities and whole-insect ATP levels were further determined during different developmental stages including 22L, 1PP, 3PP, 1-day-old pupae (1P), 3-day-old pupae (3P), and 5-day-old pupae (5P). The results showed that the in vivo AK activities was relatively stable from late larvae (22L) to late pupae (5P) (Fig. 8a), which is generally consistent with constant whole-insect ATP levels (Fig. 8b).

**Discussion**
We report on two AKs, TcAK1 and TcAK2, in T. castaneum. Amino acid alignments of their cognate proteins show the T. castaneum AKs are similar to other insects AKs. Phylogenetic analysis indicates both genes are consistent with insect AKs and clustered with group 1 and group 2 AKs, respectively. The structures of the two genes differ with a very long intron in TcAK1, but not TcAK2. The recombinant proteins demonstrate stereospecific preferences for L- over D-arginine. Injecting dsRNA

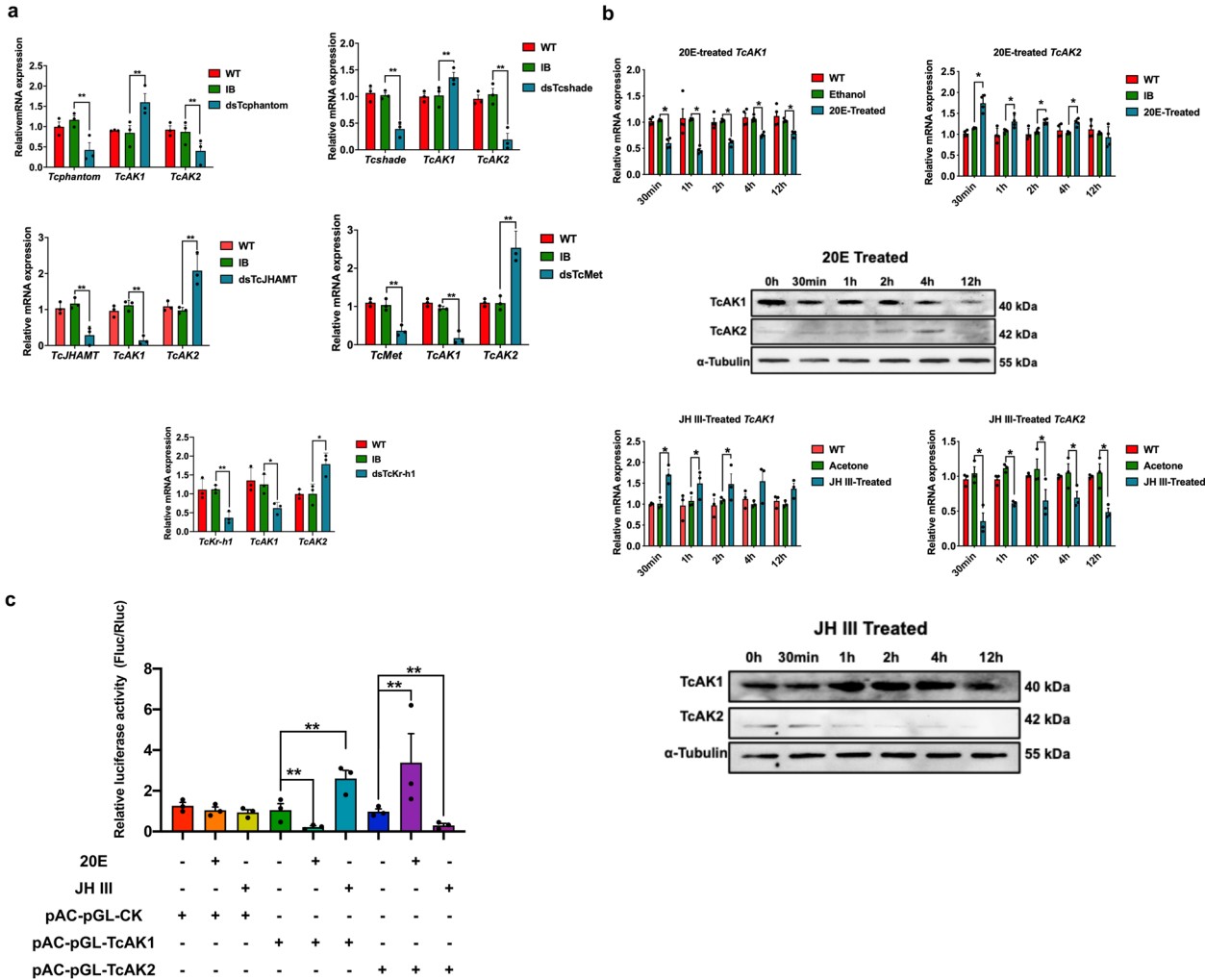

**Fig. 5 The regulation of *TcAK1* and *TcAK2* by 20E and JH III. a** The effects of knockdown of expression of *Tcphantom*, *Tcshade*, *TcJHAMT*, *TcMet*, and *TcKr-h1* on the mRNA expression of *TcAK1* and *TcAK2*. **b** The effects of treatments in vivo with 20E and JH III on the mRNA and protein expressions of *TcAK1* and *TcAK2* in larvae. **c** The effects of treatments in vitro with 20E and JH III on the promoter activity of *TcAK1* and *TcAK2* in *Drosophila* Schneider 2 (S2) cell line. These experiments were repeated at least three times with similar results. Data are expressed as mean ± SEM ($n = 3$ biologically independent replicates). Asterisks indicate differences statistically significant at **$P < 0.01$ and *$P < 0.05$ (Student's *t*-test).

leads to substantially decreased mRNAs encoding both proteins, which translated into serious development and reproductive failures. Exposure to stressors, including high and low temperatures, and the prooxidant, paraquat, led to increase in both mRNA and protein expression levels of *TcAK1* and *TcAK2*. Expression of these two genes is associated with the two major developmental hormones, ecdysteroid and JH. Taken together, each element of the research reported here constructs a characterization of the two genes and, via influence on ATP homeostasis, their biological significance in protection, reproduction, and development.

The roles of phosphagens are central in energy turnover. Tissue-specific expression analysis showed that among all examined tissues, *TcAK1* and *TcAK2* were highly expressed in head, followed by thorax. This result is consistent with fire ants, *Solenopsis invicta*, with the highest level of AK gene expression and activity in heads of female alates and workers and thorax tissue of workers[24]. Similarly, AK is highly expressed in the neuronal growth cones of locusts, *Schistocerca americana*[25] and the brain of blow flies, *Calliphora erythrocephala*[26]. These findings are consistent with the general view that the brain, neurons, and muscles have high and fluctuating energy demands.

Coincidentally, contrary to our expectations, however, the expression of *TcAK1* was significantly lower than *TcAK2* in the midgut. The ion-transporting epithelia of midgut in *Heliothis virescens*[27] and *Manduca sexta*[28] are also rich sources for AK, but the expression may well be limited to sub-sections of those tissues.

TcAK1 and TcAK2 are highly stereospecific for L-arginine versus D-arginine and showed remarkably low or no activity toward the other potential guanidine substrates. Notably, while the kinetic properties of TcAK1 are comparable to those of typical AKs, the $K_{m}^{Arg}$ value of TcAK2 is relatively high, which we take to indicate the enzyme has low substrate affinity and enzyme activity toward arginine. Like another group 2 insect AK from *C. cephalotes*[12], the catalytic efficiency of TcAK2 is also lower compared to TcAK1 and the other AKs. We speculate that TcAK1 may be the dominant AK form of the two enzymes in *T. castaneum*.

AK acts in energy-dependent processes, including development and stress responses. Exposure to dsRNA targeting *Phyllotreta striolata AK*, the beetle development, fecundity, and fertility were reduced[29]. In *Apis cerana cerana*, *AccAK* was up-regulated under stressors, including heat, cold, and prooxidants[30]. We report

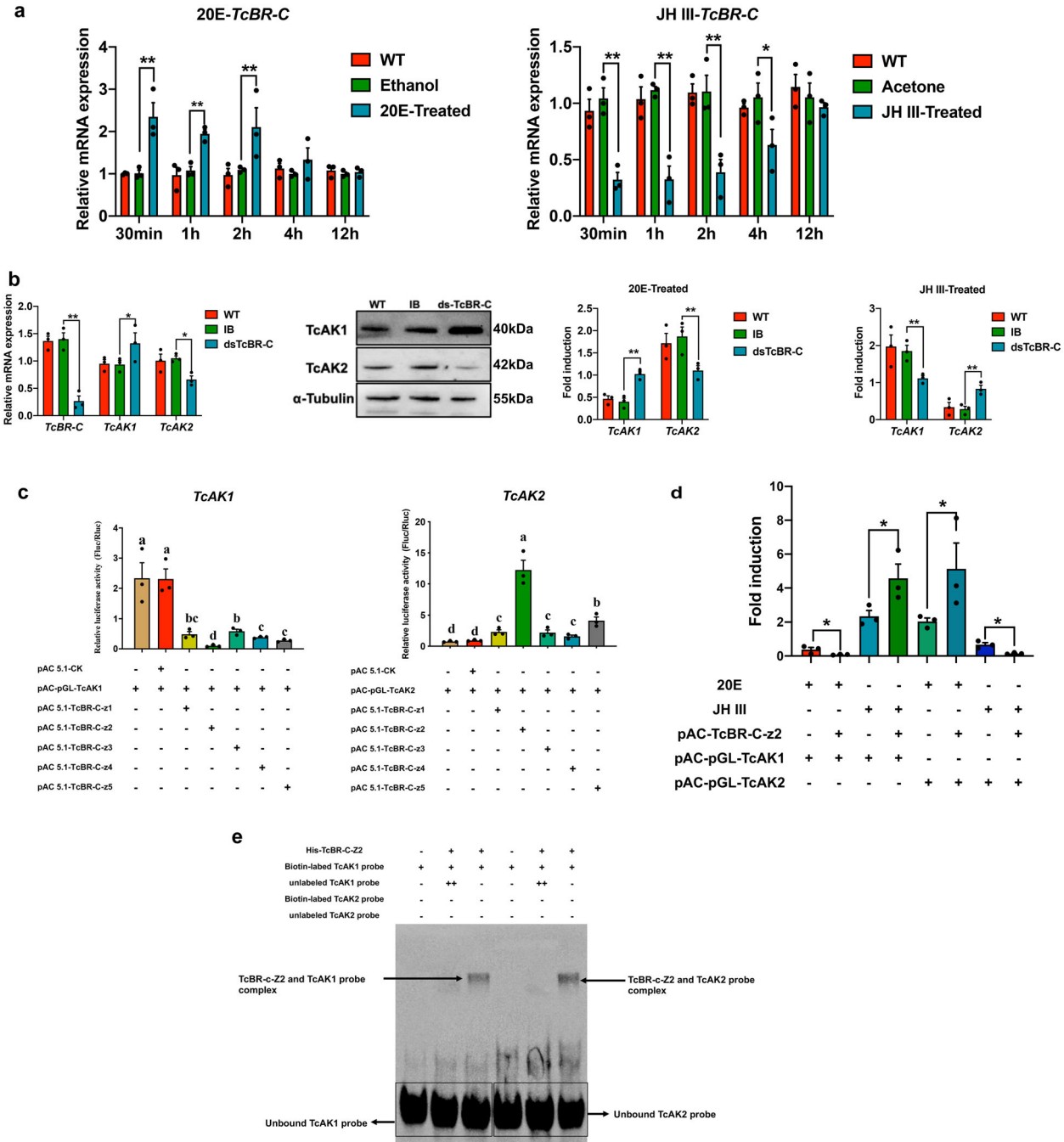

**Fig. 6 TcBR-C mediates hormonal regulation of *TcAK1* and *TcAK2*. a** The effects of treatments in vivo with 20E and JH III on the mRNA expressions of *TcBR-C* in larvae. **b** The effects of knockdown of expression of *TcBR-C* on basal expression levels and hormornal regulation of *TcAK1* and *TcAK2*. **c** The effects of overexpression of TcBR-C isoforms on *TcAK1* and *TcAK2* promoter activities in S2 cells. **d** The effects of overexpression of TcBR-C-z2 on the hormonal regulation of promoter activity of *TcAK1* and *TcAK2*. **e** EMSAs of TcBR-C-z2 binding to the promoters of *TcAK1* and *TcAK2*. Data are expressed as mean ± SEM ($n = 3$ biologically independent replicates). Histogram bars annotated with the same lowercase letters are not significantly different (one-way ANOVA, $P < 0.05$). Asterisks indicate differences statistically significant at **$P < 0.01$ and *$P < 0.05$ (Student's $t$-test).

similar findings for *T. castaneum*, as *TcAK1* and *TcAK2* were up-regulated following heat, cold, and prooxidant insults, and knockdown of both genes resulted in decreased adult fertility and stress tolerance. These may be general AK actions.

The tissue and compartment distribution of PK is a key to its functions in cellular energy metabolism. In vertebrates, compartment-specific CK isoenzymes are found in cytosol and mitochondria, which are encoded by separate nuclear genes[31–33]. In most tissues, a single cytosolic CK isoform is co-expressed together with a single mitochondrial CK isoform[34]. The ATP

production in the cytosol by glycolysis and in the mitochondria by oxidative phosphorylation are coupled with the cytosolic and mitochondria CK isoforms, respectively. Cytosolic CKs regenerate in situ ATP from the PCr pool at ATP consuming sites[1,17]. In arthropods, while AK activity associated with mitochondria has been detected in organs/tissues with continuous energy need, such as the midgut of *M. sexta*[28] and the heart of horseshoe crabs, *Limulus polyphemus*[35], AKs appear to be targeted only to the cytoplasm in *Drosophilla melanogaster*[36], and the existence of true mitochondrial AKs remains controversial[3]. In this study, we

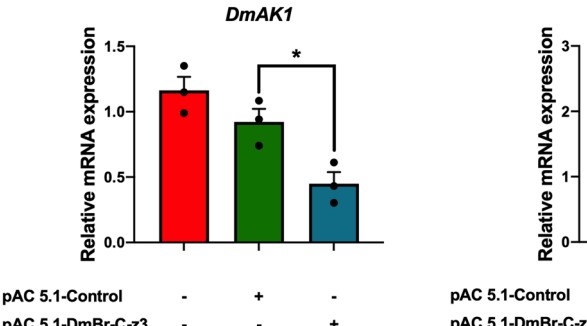

**Fig. 7 The effects of overexpression of DmBR-C-z3 on mRNA levels of two *Drosophila* arginine kinase genes, *DmAK1* and *DmAK2*, in S2 cells.** Total RNA was isolated from cells. The relative mRNA levels of *DmAK1* and *DmAK2* were determined by qPCR using *DmRp49* as an internal reference. Data are expressed as mean ± SEM ($n = 3$ biologically independent replicates). Asterisk indicates differences statistically significant at *$P < 0.05$ (Student's *t*-test).

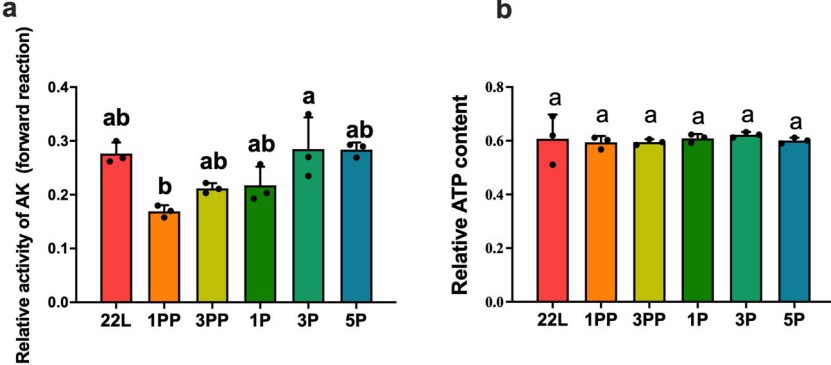

**Fig. 8 Relative AK activity and ATP content of the whole insects in different development stages. a** Relative AK activity of the whole insects in different development stages. **b** ATP content of the whole insects in different development stages. 22L 22-day-old larvae; 1PP 1-day-old prepupa; 3PP 3-day-old prepupa; 1P 1-day-old pupa; 3P 5-day-old pupa; 5P 5-day-old pupa. Data are expressed as mean ± SEM ($n = 3$ biologically independent replicates). Histogram bars annotated with the same lowercase letters are not significantly different (one-way ANOVA, $P < 0.05$).

provide evidence that TcAK1 was in cytoplasm including mitochondria, while TcAK2 was exclusively detected in cytoplasm, excluding mitochondria. This finding suggests that the typical insect group 1AKs act in cellular energy homeostasis by stimulating oxidative phosphorylation in the mitochondria and ATP consumption in the cytosol, while the group 2 AKs might mainly contribute to the energy-buffering system in the cytosol.

AKs act in ecdysone-related roles in insect development[29–38]. In *D. melanogaster*, a sharp, transient peak of AK activity occurs during the prepupal period, but not after third instar larvae of the temperature-sensitive ecdysone less mutant *ecd-1* were shifted to 29 °C[37]. In this study, after knocking down 20E and JH biosynthesis and action genes, it appeared that expression of *TcAK1* is negatively regulated by 20E and positively regulated by JH. These two hormones exhibit the opposite influences on expression of *TcAK2*, as confirmed by 20E and JH III treatments. The promoter regions of *TcAK1* and *TcAK2* were subjected to JASPAR and TFBIND analysis, and a cluster of potential BR-C binding sites were predicted in the promoter regions of both *TcAK1* and *TcAK2*. BR-C was regulated by 20E and JH[39–43] and can simultaneously regulate gene expression in a positive and negative regulatory factor[44,45]. Here, knockdown of *TcBR-C* up-regulated *TcAK1* expression whereas down-regulated *TcAK2* expression and abolishes the effects of 20E and JH III on the expression of *TcAKs*. Dual-luciferase assay and EMSA showed that TcBR-Cs can directly regulate expression of *TcAKs*. We also validated the effect of DmBR-Cs on *DmAKs* in *Drosophila* S2 cells, which was consistent with the results in the *T. castaneum*. These findings suggest to us that the opposite hormonal regulation of *TcAK1* and *TcAK2*

is mediated by TcBR-C, and the regulation pattern might be conserved in other insects with two AK genes.

The similar function and opposite hormonal regulation of *TcAK1* and *TcAK2* raised the question about the biological significances of the presence of two phylogenetically distant AK genes in several insect species. In vertebrates, the phosphocreatine-creatine kinase shuttle system is increasingly recognized as a fundamental mechanism for ATP homeostasis in both excitable and non-excitable cells, and aberrant CK levels are associated with various cancers[17,46,47]. Theoretically, in insects, the expression level of AK should also be finely regulated to ensure ATP homeostasis and cellular energy balance. The actions of ecdysteroid and JH are very well understood[48] and they are germane to our findings because the repression of *TcAK1* expression during metamorphosis could be antagonized by the up-regulation of *TcAK2*, while during juvenile–juvenile molts, the expression of *TcAK1* and *TcAK2* was adversely affected by ecdysteroid and JH. In this way, while the titers of ecdysteroid and JH in the hemolymph fluctuate in a stage-specific manner, the AK level was maintained at relatively constant level, contributing to ATP homeostasis. This hypothesis was supported by general consistency of in vivo AK activities. Further research is needed to test this hypothesis in other insect species that express two groups of AK family genes.

## Methods

**Insect strains.** The Georgia-1 (GA-1) strain of *T. castaneum* was reared on 5% (w/w) yeasted flour at 30 °C and 40% RH (relative humidity) under standard conditions[49]. The development stages of the test beetles used are as follows: 3E: 3-day-old egg; 1L:

1-day-old larvae (early first instar larval stage); 5L: 5-day-old larvae (early second instar larval stage); 20L: 20-day-old larvae (early last instar larval stage); 22L: 22-day-old larvae (mid-late last instar larval stage); 1PP: 1-day-old prepupa; 3PP: 3-day-old prepupa; 1P: 1- day-old pupa; 3P: 3-day-old pupa; 5P: 5-day-old pupa; 1F: 1-day-old female adult; 7F: 7-day-old female adult; 1M: 1-day-old male adult; 7M: 7-day-old male adult.

**Total RNA isolation and reverse transcription**. Total RNAs were extracted from the whole bodies or specific tissues using the SV total RNA isolation system (Promega, Madison, WI). First-strand cDNA was synthesized from total RNA using the Primescript™ First-Strand cDNA Synthesis kit (TaKaRa, Dalian, China), following the manufacturer's instructions.

**Polymerase chain reaction and RACE**. The amino acid sequence of a *D. melanogaster* AK (GenBank: NM 079264) was searched against BeetleBase (http://www.bioinformatics.ksu.edu/blast/bblast.html), and the regions with significant hits were manually annotated to identify the transcripts. Specific primer pairs were designed based on the identified sequences (Supplementary Table 1). PCR reactions were performed with LA Taq™ DNA polymerase (TaKaRa, Dalian, China). To complete the cDNA sequences of *TcAK1* and *TcAK2*, 5′-RACE and 3′-RACE reactions were performed using the SMART RACE cDNA Amplification Kit (Clontech, Mountain View, CA, USA), following the manufacturer's instructions.

**qPCR**. qPCR reactions were performed on the Bio-Rad CFX 96 Real-time PCR system using SYBR®PrimeScript™ RT-PCR Kit II (Takara, Dalian, China) and gene-specific primers (Supplementary Table 1). The efficiency of primer pairs was validated before gene expression analysis. The stably expressed gene encoding ribosomal protein S3 (rps3, GenBank: CB335975) was used as a reference gene[50]. The mRNA levels were normalized to reference gene with the $2^{-\Delta\Delta CT}$ method[51]. The means and standard errors for each time point were obtained from the average of at least three biologically independent sample sets.

**In vitro expression and purification**. Based on the deduced start and stop codon positions in the 5′- and 3′-RACE product sequences, primers were designed for the RT-PCR amplification of the ORF of both genes with *Kpn*I and *Xma*I restriction sites introduced into forward and reverse primer sequences (Supplementary Table 1). The PCR product was cloned into the pMD18-T vector (TaKaRa, Dalian, China), excised by *Kpn*I/*Xma*I (NEB, Ipswich, MA, USA) digests and subcloned into *Kpn*I/*Xma*I-cut pQE-30-XA. Recombinant *TcAK1* and *TcAK2* were confirmed by double enzyme digestion and sequencing. The correct recombinant product was transformed into *Escherichia coli* M15. Purification of recombinant TcAK1 or TcAK2 was performed using native $Ni^{2+}$-NTA agarose chromatography as outlined by the manufacturer (Qiagen, Valencia, CA, USA). The enriched recombinant proteins were assayed by SDS-PAGE and confirmed by western blot with an anti-6×His-Tag monoclonal antibody (Invitrogen, Carlsbad, CA, USA, MA1-21315, 1:8000), following the manufacturer's instructions.

**In vitro enzyme assays**. BCA Protein Assay Kit (Thermo Scientific Pierce, IL, USA) was used to determine the protein concentration in column eluates according to the manufacturer's protocol. AK assays were performed as described previously[52] and according to Sigma's product instruction manual (Product No. A3389). Briefly, in forward reaction (phosphagen synthesis), a 0.2 mL reaction mixture was prepared by combining 50 mM Tris-HCl, pH 7.5, 20 mM $MgCl_2$, 0.9 mM ATP, 0.2 mM NADH, 2.5 mM phosphoenolpyruvate, 4 mM DTT, 2 units/mL pyruvate kinase and 1 unit/mL lactate dehydrogenase. After adding recombinant enzyme to the reaction mixture, the reaction was started by the addition of L-arginine to a final concentration of 5 mM. The Michaelis–Menten constant ($K_m$) for L-arginine was determined based on varying the concentrations of L-arginine between 0.2 and 10 mM. To determine the $K_m$ values for phospho-L-arginine, the reaction mixture contained 50 mM Tris-HCl pH 7.5, 20 mM $MgCl_2$, 10 mM glucose, 2 mM nicotinamide adenine dinucleotide phosphate ($NADP^+$), 4 mM DTT, 2 units/mL hexokinase, and 1 unit/mL glucose-6-phosphate dehydrogenase. The initial velocity values were obtained by varying the concentration of phospho-L-arginine between 0.05 and 5 mM under the fixed concentrations of 5 mM ADP. All reactions were carried out four times at 30 °C. Absorbance was measured at 340 nm estimate kinetic constants ($K_m$ and $k_{cat}$), a Lineweaver–Burk plot was made and fitted by the least-square method in Microsoft Excel.

**Sequence analysis and database entries**. Sequence alignment was performed using ClustalW[53] with the default settings. The aligned sequences were used for construction of the phylogenetic tree in MEGA5 with 1000 bootstrap replicates[54]. The entire coding sequences of *TcAK1* and *TcAK2* have been deposited in the GenBank and the accession numbers are KY971526 and KY971527, respectively.

**RNA interference**. Double-stranded RNAs (dsRNAs) were synthesized using TranscriptAid™ T7 High Yield Transcription Kit (Thermo Fisher Scientific, Waltham, MA, USA) based on nucleotides 779–1042 (264 bp) and 833–1066 (234 bp) of the ORF regions of the *TcAK1* and *TcAK2*, respectively. About 200 ng of dsRNA in 200 nL solution was injected into 20L using a Nanoliter 2010 injector system (WPI, Sarasota, FL, USA). On the 4 days post dsRNA injection, the insects were used to determine the suppression of *TcAK1* and *TcAK2* transcripts by qPCR as described for gene expression analysis. The experimental insects were reared under the standard conditions mentioned above, and the phenotypes were observed. The buffer-injected larvae (IB group) and the uninjected wild-type larvae (WT group) were set as controls in all injection experiments. Three biologically independent replicates were carried out with at least 30 insects in each replicate.

Because dsTcAK1 treatments in larvae induced significant mortality, pupae were used for RNAi and about 200 ng of dsTcAK1 or dsTcAK2 in 200 nL solution was injected into 2-day-old pupae. After adult eclosion, they were transferred to fresh Petri dishes with food at 30 °C, and target gene suppression was determined in 4-day-old female adults. Pairs consisting of a virgin 10-day-old female and a male were crossed and transferred to fresh vials. The fecundity, as numbers of eggs laid, and fertility, as egg hatchability, were recorded. Three biologically independent replicates were performed, with 25 pairs/replicate.

**Stress assays**. We conducted two stress assays, a stress-induced gene expression assay and a stress tolerance assay. In the first, two-day-old females were divided into several groups and placed under various environmental stresses for *TcAK1* and *TcAK2* expression analysis. In the starvation-treatment group, the adult beetles were kept at 30 °C without food and collected every 2 days (3/collection) after treatment until death. In the heat (45 °C) and cold (4 °C) treatment groups, 50 females for each group were maintained for 0, 1, 2, 4, and 12 h, and 3 females were collected from each time point. In the oxidative stress group, adults were starved for 12 h and then treated with 20 mM paraquat. Three insects were collected at the time points after insects were treated for 0, 1, 2, 4, and 12 h, respectively. In total, three independent replicates were carried out for these experiments. The mRNA expression levels of *TcAK1* and *TcAK2* were determined by qPCR.

For western blot, a 14-amino acid peptide derived from the C-terminal end of TcAK1 (NDIEKRLPFSHSDR) and a 14-amino acid peptide derived from the C-terminal end of TcAK2 (EEIETKLKFSRSDR) with an additional N-terminal Cys residue was synthesized, conjugated with keyhole limpet hemocyanin, and used as antigen to inject rabbits for antibody production. All antibody concentrations were diluted to 1 mg$^{-1}$ mL$^{-1}$. All procedures were conducted by Genscript Corporation (Nanjing, China). Antibody specificity was tested using bacterially expressed recombinant TcAK1 and TcAK2 as described above[55] (Supplementary Fig. 12). Total protein was extracted from whole insects of the stress treated and control groups using a Tissue Protein Extraction Kit (ComWin Biotech, Beijing, China), and quantified using a BCA Protein Assay Kit (Thermo Scientific Pierce, IL, USA) according to the manufacturer's protocol. The western blot was conducted separately for TcAK1 and TcAK2 with the same protein samples extracted from different treatment groups. Briefly, equal amounts of protein (50 μg) were separated on 10% SDS-PAGE, and then transferred to PVDF with transfer buffer composed of 25 mM Tris-HCl (pH 8.9), 192 mM glycine, and 20% methanol. Membranes were blocked by 5% (w/v) bovine serum albumin (BSA) in Tris-buffered saline with 0.05% Tween-20 (TBST) for 1 h at room temperature and then incubated overnight at 4 °C with specific polyclonal antibodies against TcAK1 and TcAK2 of 1:1000 dilution in 5% BSA in TBST and α-Tubulin mouse monoclonal antibody (Proteintech Group, Inc., Chicago, IL, USA, 02502, 1:5000) was used as a loading control. This was followed by incubation with HRP-coupled secondary anti-rabbit or anti-mouse antibodies (Proteintech Group, Inc., Chicago, IL, USA), respectively. The blot signals were detected using Tanon™ High-sig ECL Western Blotting kit (Tanon, Shanghai, China) with a ChemiDoc™ Touch Imaging System (Bio-Rad, Hercules, CA, USA). The relative expression levels of proteins were quantified densitometrically using the software Image Lab (Bio-Rad, Hercules, CA, USA) and calculated according to the reference bands of α-tubulin.

The stress tolerance treatments were the same as those described in the stress-induced expression assay, except that insects treated with dsRNA were used. The survival rates were counted every 12 h during starvation, every 6 h up to 48 h for thermal and cold stress treatments, and every 3 h up to 24 h for the paraquat treatments. The IB group and WT group were set as controls. Three biological replications were carried out for the experiments, and at least 50 beetles were used in each replicate treatment.

**Subcellular localization**. Dissected midgut was washed three times with PBS. For imaging of mitochondria, midguts were incubated with 250 nM MitoTracker Red CMXRos (Beyotime, Nantong, China) in PBS for 30 min at 37 °C, and excess dye was washed off with PBS, followed by fixation with 4% (v/v) ice-cold paraformaldehyde in PBS for 2 h, washed three times with PBS and permeabilized with PBS containing 1% (v/v) Triton X-100 for 1 h at room temperature. Nonspecific antibody-binding sites were blocked by incubating with 5% BSA in PBS containing 0.1% (v/v) Tween-20 (PBST) for 1 h at 37 °C. For immunostaining, tissues were incubated overnight at 4 °C with anti-TcAKs antibodies as described above. After washing with PBS, midguts were then incubated for 1 h with Alexa Fluor 488 dye (Invitrogen, Carlsbad, CA, USA) diluted at 1:1000 with PBST. After washing with PBST, tissues were mounted on a microslide with the DAPI mounting medium (Solarbio, Beijing, China) to mount slides (Vector Laboratories, Burlingame, CA, USA) to visualize the cell nucleus. Images were acquired with a Leica TCS SP8

STED 3X super-resolution microscope and processed by using Leica Application Suite Advanced Fluorescence software (LAS AF, Leica, Heidelberg, Germany).

To confirm the immunofluorescence staining result, nuclear, mitochondrial, and cytoplasmic proteins were extracted with the Nuclear and Cytoplasmic Protein Extraction Kit and Tissue Mitochondria Isolation Kit (Beyotime Institute of Biotechnology, Nantong, China) from 30 adults according to the manufacturer's instructions. Western blot analysis was conducted as described above, and anti-α-Tubulin antibody, anti-COX IV antibody (Abcam, Cambridge, UK, ab14744, 1:5000), and anti-Histone H3 antibody (Abcam, Cambridge, UK, ab1791, 1:5000) were used as loading controls for cytoplasmic, mitochondrial, and nuclear fraction, respectively.

**Regulation assays.** To test the hormonal regulation of *TcAK1* and *TcAK2*, dsRNAs targeting two ecdysteroid biosynthesis genes, *Tcphantom*, *Tcshade* and three JH biosynthesis and action genes, *TcJHAMT*, *TcMet* and *TcKr-h1*, were synthesized using primer pairs as previously reported[56,57] (Supplementary Table 1) and injected into 20L. On the fourth day after dsRNA injection, the insects were used to determine relative expression of the target genes and *TcAK1* and *TcAK2* transcripts by qPCR.

The hormonal regulation of *TcAK1* and *TcAK2* was confirmed by hormonal treatments in vivo and in vitro. Generally, JH III (Sigma-Aldrich, St. Louis, MO, USA) was applied topically (5.0 μg/μL in acetone, 200 nL/insect) on the dorsum of 20L. A stock solution (10 μg/μL) of 20E (Sigma-Aldrich, St. Louis, MO) dissolved in 95% ethanol was diluted to 1 μg/μL with distilled water, and the 20L were injected with 0.3 μL of 20E solution. Insects were collected at 0.5, 1, 2, 4, and 12 h after treatment. The same volume of solvent was applied as a control. The mRNA and protein expression levels were determined by qPCR and western blot as described above, respectively.

In vitro testing was conducted in *Drosophila* Schneider 2 (S2) cells using dual-luciferase reporter system. Reporter vector pAC-pGL3.1 carrying *Drosophila* actin 5c (Ac5) promoter (kindly gifted by Professor Bin Li at Nanjing Normal University, China) and overexpression vector pAC5.1 (Invitrogen, Carlsbad, CA) were linearized by double restriction enzyme digestion with restriction enzyme *Kpn*I and *Nhe*I, and *Bam*HI and *Not*I (NEB, Ipswich, MA, USA), respectively. The genomic DNA was extracted using MiniBEST Universal Genomic DNA Extraction Kit Ver.5.0 (TaKaRa, Dalian, China) according to the manufacturer's instructions. The upstream regulatory regions of *TcAK1* (from −2170 to +130 bp) and *TcAK2* (from −2079 to +88 bp) were amplified from genomic DNA using primers sharing 15 homologous bases at each end of linearized vector pAC-pGL3.1 (shown in Supplementary Table 1). The PCR products were cloned into the vector pAC-pGL3.1. The In-Fusion reaction was performed according to the instruction of In-Fusion® HD cloning kit (Clontech, CA, USA). Briefly, the reaction mixture contained 2.0 μL of 5× In-Fusion HD Enzyme Premix, 100 ng linearized vector, 200 ng of each purified PCR fragment, and sufficient ddH₂O to adjust the reaction to a total volume of 10 μL. The reaction mix was incubated at 50 °C for 15 min, and then placed on ice for transformation using *E. coli* DH5α Premium Competent Cells (TaKaRa, Dalian, China). All constructs were sequenced to ensure the completeness of the constructs. S2 cells were seeded at a density of $1.5 \times 10^5$ cells/well in 200 μL Schneider's *Drosophila* Medium containing 10% fetal bovine serum in a 96-well plate (Sumilon, Sumilomo Bakelite Co., Ltd, Tokyo, Japan) 1 day before transfection. Transfection of *Drosophila* S2 cells was performed using FuGENE HD (Promega, Madison, WI, USA). Briefly, 1 μg pAC-pGL-TcAK1 or pAC-pGL-TcAK1 transfection into the cells and 0.1 μg pIZT-RLuc vector containing the *Renilla* luciferase gene constructed as previously described[58] was used as the reference for insect cells. The transfection of pAC-pGL-CK vector was used as control. After 24 h, the medium was replaced with fresh one containing 1 μM 20E or 250 ng/ml JH III. Luciferase activity was assayed at 4 h after JH III treatment and 24 h after 20E treatment, respectively, using Dual-Luciferase reporter assay system (Promega, Madison, WI, USA) according to the protocol and analyzed with a Synergy HT fluorometer (BioTek Instruments, Winooski, VT, USA). Transfection was repeated three times ($n = 3$) and the average expression levels of the report genes were represented as mean ± SEM.

The search of JASPAR database (http://jaspar.genereg.net/) and TFBIND (http://tfbind.hgc.jp/) identified BR-C as potential transcription factors regulating *TcAKs* expression. Given that five BR-C isoforms have been identified in *T. castaneum*, dsRNA targeting common region of *TcBR-C* was synthesized as previously described[59] (primers shown in Supplementary Table 1) and injected into the 15L. On the fourth day after injection, RNAi efficiency was determined, and the mRNA and protein levels of TcAK1 and TcAK2 were detected. dsRNA-injected and control larvae were treated with 20E or JH III for 2 h using the methods as described above. The mRNA expression levels of *TcAK1* and *TcAK2* were detected by qPCR.

For dual-luciferase reporter assay, reporter vectors were constructed as described above. The ORFs of five isoforms of *TcBR-Cs* were amplified from the cDNA of the 1-day-old adults using primers with 15 bp extensions complementary to the ends of the desired linearized pAC5.1 vector cloning site for homologous recombination (shown in Supplementary Table 1). The RT-PCR products were cloned into the vector pAC5.1, respectively. Co-transfection was performed as described above. One microgram pAC-pGL-TcAK1/pAC-pGL-TcAK2 and 0.5 μg pAC-TcBR-Cs were co-transfection into the cells and 0.1 μg pIZT-RLuc vector was

used as the reference for insect cells. Luciferase activities were assayed at 48 h post transfection as described above. Co-transfection was repeated three times ($n = 3$) and the average expression levels of the report genes were represented as mean ± SEM.

EMSAs were used to assess DNA-binding activity of TcBR-C on the *TcAK1* and *TcAK2* promoters. The probe for *TcAK1* (5′-CCGGGAGAGGGGATTCTGCAAATAGATTTTTATCAGTTAATTCCTT-3′) and probe for TcAK2 (5′-TCTCCCTTATTTATTGAAATATAAATAGAATTCTGAAATT-3′) were synthesized and labeled with biotin using an EMSA Probe Biotin Labeling Kit (Beyotime, Nantong, China) according to the manufacturer's instructions. The underlined nucleotides denote the binding sites of BR-C. The full-length coding region of TcBR-C-z2 in pAC-TcBR-C-z2 was subcloned into the expression vector pGEX4T-1 to generate the GST-TcBR-C-z2 fusion vector as described above. Primers sequence for protein expression are shown in Supplementary Table 1. GST-TcBR-C-z2 recombinant protein was purified using Mag-Beads GST Fusion Protein Purification (Sangon, Shanghai, China) according to the manufacturer's instructions and used for the EMSA. The enriched recombinant protein was assayed by SDS-PAGE and confirmed by western blot with anti-GST-Tag monoclonal antibody (Proteintech Group, Inc., Chicago, IL, USA, 10000-0-AP, 1:8000) (Supplementary Fig. 13). In competition experiments, unlabeled probes were added to the reactions. EMSA was performed using a chemiluminescent EMSA kit (Beyotime, Nantong, China) according to the manufacturer's instructions.

Expression plasmids of DmBR-C-z1, DmBR-C-z2, DmBR-C-z3, and DmBR-C-z4 (pAC- DmBR-C-z1, z2, z3, and z4) were constructed as describe above. The full ORFs of DmBR-C-z1, z2, z3, and z4 were amplified by PCR from full-length cDNA using the primers sharing 15 homologous bases at each end of linearized vector pAC-3.1 listed in Supplementary Table 1. The overexpression vector was transfected into S2 cells separately as described above, and after 48 h of treatment, the effects of *DmAK1* and *DmAK2* expression using qPCR and Rp49 were used as an internal reference gene[60]. The primers used to detect the expression of *DmAK1* and *DmAK2* are listed in Supplementary Table 1.

**In vivo assays of AK activities and ATP levels.** Assay of AK activity was conducted according to Zhao et al.[61] and Sigma's product instruction manual (Product No. A3389). Briefly, 30 individuals of different developmental stages including 22L, 1PP, 3PP, 1P, 3P, and 5P were collected and washed with PBS for three times. The beetles were pestled in elution buffer (0.25 M sucrose, 0.05 M Tris-HCL, 1 mM EDTA, pH 7.4), and homogenates were centrifuged at 12,000 r.p.m. for 30 min at 4 °C to obtain crude enzyme solution. Protein concentration was determined as described above. AK activity was assayed according to Zhao et al.[61] and Sigma's product instruction manual (Product No. A3389). To detect hormonal modulation of ATP levels, we treated 20L with 20E and JH III according to the method described above. As for ATP analysis, 30 beetles were homogenized in 200 μL lysis buffer. The homogenates were centrifuged at $12,000 \times g$ for 5 min at 4 °C. The supernatant was transferred to a new 1.5-mL tube, and the ATP content was measured by using an ATP assay kit (Beyotime, Shanghai, China) based on a bioluminescence technique according to the manufacturer's instructions. Protein concentration was measured using the method described above. The relative ATP level was expressed as ATP value/protein value[62].

**Statistical analysis and reproducibility.** Data are presented as mean values ± SEM. Student's *t*-test was used for comparing two means and ANOVA with post hoc Tukey's HSD test was used for multiple comparisons of parametric data. All statistical analyses were performed using SPSS software (SPSS 13.0 for windows; SPSS Inc., Chicago, IL, USA). Differences were considered statistically significant when $P < 0.01$ (**) or $P < 0.05$ (*). All experiments were conducted in at least three independent replicates. The sample size is indicated for each experiment in the corresponding figure legend.

**Reporting summary.** Further information on research design is available in the Nature Research Reporting Summary linked to this article.

## Data availability

Uncropped blots are shown in Supplementary Fig. 14. All other relevant data are included in the paper and its Supplementary information. The entire coding sequences of *TcAK1* and *TcAK2* have been deposited in the GenBank and the accession numbers are KY971526 and KY971527, respectively.

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

## Acknowledgements

This work was supported by the National Natural Science Foundation of China under grant nos. 31572000 and 31871974. Mention of trade names or commercial products in this article is solely for the purpose of providing specific information and does not imply recommendation or endorsement by the U.S. Department of Agriculture. All programs and services of the U.S. Department of Agriculture are offered on a nondiscriminatory basis without regard to race, color, national origin, religion, sex, age, marital status, or handicap.

## Author contributions

J.W. conceived and designed the experiments. N.Z., H.J., X.M., and Y.L. performed the experiments. J.W. and Y.P. supervised the work. H.J. and K.Q. analyzed the data. J.W., N.Z., D.S., Q.S., and I.W. wrote the paper.

## Competing interests

The authors declare no competing interests.
