## [Peer Review File · Communications Biology]

Reviewers' comments:

Reviewer #1 (Remarks to the Author):

The manuscript characterizes two phylogenetically distant AK genes and explores opposing hormonal regulation of the two AK genes in *Tribolium castaneum*. This work is comparatively systematical. Here, the authors propose a model that TcAK1 is negatively regulated by 20E and positively by JH, while TcAK2 is regulated by the opposite pattern, during metamorphosis, which maintains the ATP homeostasis in insects. Although an interesting model is presented, key questions need to be answered and additional experiments should be performed regarding this work. Also, I think this article is suitable to be published in a specialized journal about insect study.

1.The novelty of this work is establishing the relationship between two antergic hormones and the ATP homeostasis which is regulated by AK genes. However, this work is much more redundant in introducing the basic knowledge of the two AK genes which are mainly explored in other species. So, I think that Figure1, figure3, figure6, figure7, figure8 and figure9 are not necessary to presentation.

2.In figure8, the mitochondria-specific markers MitoTracker Red CMXRos that exhibit the location of TcAK1 and TcAK2 are comparatively different. Does there has something wrong in the staining? It is needed to be stained again.

3.In figure 12, western blot assays should be carried out to test the effects of BR-C on the expression of TcAK1 and TcAK2.

4.Whether BR-C overexpression in cell line can also influence the expression of AK1 and AK2.

5.In figure11 and figure12, the ATP content should to be calculated under the treatment of the hormones and BR-C RNAi.

6.As is known, Kr-h1 transcription factor is the key regulator that mediates the repressive action of JH on insect metamorphosis and inhibits the expression of BR-C, so the authors need to characterize the function of Kr-h1 on TcAK1 and TcAK2.

7.Whether several potential Kr-h1 binding sites are predicted on the promoters of TcAK1 and TcAK2?

8.In figure13, the authors just performed dual-luciferase assay to verify the regulation of BR-C on TcAK1 and TcAK2. Muchmore methods ought to be used, such as Chip-PCR and EMSA, et al.

9.The figures in this manuscript are always piecemeal. It is necessary to renew and integrate the figures.

10.More recent references regarding to insect BR-C and hormones should be cited.

Reviewer #2 (Remarks to the Author):

The manuscript by Zhang et al. deals with the characterization of two arginine kinase genes in the red flour beetle *Tribolium castaneum*. The authors first identify two phylogenetically distant AK genes, AK1 and AK2 and show that AK1 might be the dominant form during the life cycle of the beetle. Next, by using a RNAi approach, the authors functionally characterize both genes in the context of metamorphosis and adulthood. They show that AK1 is required for metamorphosis and

that AK1 and AK2 are necessary in the adult stage for proper fertility and for the stress response. They also characterize the different subcellular distribution of both enzymes and also analyze the complex hormonal regulation of AK1 and AK2 expression during larval development by 20-hydroxyecdysone and juvenile hormone.

The paper is well written and, in general, the experiments are properly designed and conducted appropriately. However, the work presents a number of problems that must be carefully addressed before the manuscript can be considered for publication:

1. The developmental expression of both genes during larval development as well as the RNAi analysis conducted on larvae are very difficult to interpret given that larvae are staged by days instead of instar stages. What does 1L, 5L and 22L mean? Given that *Tribolium* larvae undergo 7-8 larval instars and that in each instar there are periods of low and high 20E titers and presence and absence of JH, it is very difficult to interpret the developmental expression levels by merely stating how old are the larvae. Same applies to the RNAi experiments. The authors claim that the dsRNAs are injected in 20-day-old larvae and that mRNA levels for AK1 and AK2 are measured 4 days later. What does 20 days-old mean? Are these larvae in the penultimate larval instar? Are they in the last larval instar? This point is very important to understand the corresponding phenotypes. I suggest to clarify the staging of the samples and different injections throughout the paper by clearly describing in which instar are all the treated animals.

2. The comparison between AK1 and AK2 mRNA abundance throughout development is based on qPCR, which depends on the efficiency of the primers used in each case. To be sure that the differences are real and can be compared, it is important to check that the efficiency of both primer sets are optimal. If the authors have analyzed the efficiency of both, AK1 and AK3 primers, with a standard curve then it must be stated in the corresponding section of material and methods section. If not, the authors must run this procedure.

3. The morphological description of larval RNAi phenotype for AK1 is absent. Although clearly affected, it is very difficult to see the morphological defects of AK1-RNAi animals in Fig. 4. A more detailed description of the abnormalities would be necessary. Better and more clear pictures can also help to visualize the phenotype.

4. Regarding the western blot analysis of AK1 and AK2, my main concern is to clarify how these western were obtained. In material and methods, the authors stated that "membranes were incubated with antiAK1 and antiAK2", thus suggesting that both antibodies were incubated simultaneously. If this is the case, and given that both proteins have almost the same molecular weight and the relative position in the membrane of both is almost identical (see Figure S4), it is strange to see that bands for AK1 and AK2 in the western blots (Figs 6, 9 and 11) are very clean and well separated from each other. Given that both bands run at almost the same position, it is very difficult to think that the membranes have been cut before incubating them with the different antibodies. Therefore, to clarify this point, the authors might show the whole membranes for each western blot analysis. On the contrary, if different western blots were carried out for each AK proteins, two different anti-tubulin blots are required for each AK protein western blot.

5. Regarding the Br-C-RNAi experiments, some clarifications are required to understand their significance. First, and as stated in point 1, a clear description of the larval stage used is required. This is particularly important in this case as Br-C expression is stage specific with a restricted strong pulse of expression in mid-late last larval instar. Second, expression levels of AK1 and AK2 in dsTcBr-C (without 20E or JH) must be measured as it is important to see how the absence of Br-C affects the expression of both AK genes. Third, and given that Br-C expression is highly affected by the presence of 20E and/or JH as the authors claim in the text, it must be reasoned that the administration of 20E and JH in wt or IB larvae already affects the levels of Br-C in these animals, thus affecting the expression of AK1 and AK2. Therefore, to take this point into consideration, the levels of Br-C must be measured and reported in all treatments presented in Fig

12B. Likewise, the authors can also measure the levels of Kr-h1 as a readout of JH-dependent gene in all cases. Knowing the levels of Br-C and Kr-h1 can clarify the response of AK1 and AK2 to 20E and JH.

6. Regarding the cell transfection experiments (Fig. 12). Given that Br-C can activate or repress the expression of AK1 and AK2 in the presence/absence of 20E and JH, it would be interesting to mimic the in vivo experiments reported in Fig. 11 in the cell transfection experiments. First, S2 cells transfected with pAC-pGL-TcAK1 or pAC-pGL-TcAK12 must be incubated with 20E and JH alone to measure the expression of AK1 and AK2. Second, S2 cells transfected with pAC-pGL-TcAK1 or pAC-pGL-TcAK12 and PAC5.1-TcBr-C-Z2 (ideally, with all different isoforms) must be incubated with 20E and JH to see if the activation of the promoter behaves as in the in vivo experiments.

Moreover, statistic differences between different PAC5.1-TcBr-C-Z constructs can no be taking into consideration if the levels of each transfected protein are not measured by western blot to normalize the effect by the amount of protein produced.

Comments:

The manuscript characterizes two phylogenetically distant AK genes and explores opposing hormonal regulation of the two AK genes in *Tribolium castaneum*. This work is comparatively systematical. Here, the authors propose a model that TcAK1 is negatively regulated by 20E and positively by JH, while TcAK2 is regulated by the opposite pattern, during metamorphosis, which maintains the ATP homeostasis in insects. Although an interesting model is presented, key questions need to be answered and additional experiments should be performed regarding this work. Also, I think this article is suitable to be published in a specialized journal about insect study.

1. The novelty of this work is establishing the relationship between two anergic hormones and the ATP homeostasis which is regulated by AK genes. However, this work is much more redundant in introducing the basic knowledge of the two AK genes which are mainly explored in other species. So, I think that Figure1, figure3, figure6, figure7, figure8 and figure9 are not necessary to presentation.
2. In figure8, the mitochondria-specific markers MitoTracker Red CMXRos that exhibit the location of TcAK1 and TcAK2 are comparatively different. Does there has something wrong in the staining? It is needed to be stained again.
3. In figure 12, western blot assays should be carried out to test the effects of BR-C on the expression of TcAK1 and TcAK2.
4. Whether *BR-C* overexpression in cell line can also influence the expression of AK1 and AK2.
5. In figure11 and figure12, the ATP content should to be calculated under the treatment of the hormones and *BR-C* RNAi.
6. As is known, Kr-h1 transcription factor is the key regulator that mediates the repressive action of JH on insect metamorphosis and inhibits the expression of BR-C, so the authors need to characterize the function of Kr-h1 on TcAK1 and TcAK2.
7. Whether several potential Kr-h1 binding sites are predicted on the promoters of *TcAK1* and *TcAK2*?
8. In figure13, the authors just performed dual-luciferase assay to verify the regulation of BR-C on TcAK1 and TcAK2. Muchmore methods ought to be used, such as Chip-PCR and EMSA, et al.
9. The figures in this manuscript are always piecemeal. It is necessary to renew and integrate the figures.

10. More recent references regarding to insect BR-C and hormones should be cited.

Dear Sir/Madam:

We are grateful for all insightful comments from the editor and reviewer. The manuscript (COMMSBIO-19-1718-T) has been carefully revised according to your insightful comments and suggestions. We hope that the changes having been made to the manuscript meet to your satisfaction. The following is our point-to-point response.

Best Regards,

Jianjun Wang

Editor's comments:

We hope you will find the referees' comments useful as you decide how to proceed. Should further experimental data or analysis allow you to address these criticisms, we would be happy to look at a substantially revised manuscript. However, please bear in mind that we will be reluctant to approach the referees again in the absence of major revisions, which I summarized below:

1) address the discrepancy for results in Fig8 and/or repeat the staining (Reviewer 1).

Reply:

We have explained this discrepancy in our response to Reviewer 1.

2) do western blots to test the effect of BR-C on TcAK1/2 expressions (Reviewer 1) and address the point about the Western Blots for Figs 6, 9 and 11 (Reviewer 2).

Reply:

We have conducted western blots suggested by Reviewer 1 and addressed the points in our response to Reviewer 2.

3) test the effect of BR-C over-expression in cell line (Reviewer 1).

Reply:

We have conducted related experiments suggested by Reviewer 1.

4) calculate the ATP content in response to hormone treatment and BR-C RNAi (Reviewer 1).

Reply:

We have conducted related experiments suggested by Reviewer 1.

5) characterize the function of Kr-h1 on TcAK1/2 and predict the former's binding sites to the latter's promoters (Reviewer 1).

Reply:

We have conducted related experiments suggested by Reviewer 1 and addressed the points in our response to Reviewer 1.

6) confirm the regulation of BR-C on TcAK1/2 additionally with ChIP-PCR or EMSA (Reviewer 1).

Reply:

We have conducted EMSA suggested by Reviewer 1.

7) clarify the staging of the samples and the different injections (Reviewer 2).

Reply:

We have addressed the points in our response to Reviewer 2.

8) check the primer efficiency (Reviewer 2).

Reply:

We have addressed the point in our response to Reviewer 2.

9) provide description of the phenotypic effects of RNAi in Fig4 (Reviewer 2).

Reply:

We have addressed the point in our response to Reviewer 2.

10) measure the expressions of AK1 and 2 in dsTcBr-C (without 20E of JH) (Reviewer 2).

Reply:

We have conducted related experiments suggested by Reviewer 2.

11) measure Br-C in Fig12B as well as Kr-h1 as a read-out of JH-dependent genes (Reviewer 2).

Reply:

We have conducted Br-C-related experiments suggested by Reviewer 2 and addressed the points of Kr-h1 in our response to Reviewer 2.

12) normalize the western blot by the amount of protein produced (for the levels of the transfected protein) (Reviewer 2).

Reply:

We have addressed the point in our response to Reviewer 2.

13) Address textual/method/figure clarifications brought up by the reviewers.

Reply:

We have revised the manuscript according to reviewer's comments and suggestions.

Reviewer #1:

1.The novelty of this work is establishing the relationship between two antergic hormones and the ATP homeostasis which is regulated by AK genes. However, this work is much more redundant in introducing the basic knowledge of the two AK genes which are mainly explored in other species. So, I think that Figure1, figure3, figure6, figure7, figure8 and figure9 are not necessary to presentation.

Reply:

To the best of our knowledge, there is no comparative characterization of two phylogenetically distant arginine kinase genes in insects. According to reviewer's suggestion, Figures 1, 3, 6, 7 have been moved to supplementary material, but we still retained Figure. 8 and Figure 9 (Figure 4 in the revised manuscript).

2. In figure8, the mitochondria-specific markers MitoTracker Red CMXRos that exhibit the location of TcAK1 and TcAK2 are comparatively different. Does there has something wrong in the staining? It is needed to be stained again.

Reply:

We are grateful for this comment. Initially, tissues were prepared for immunofluorescence staining with two methods including cryosectioning and dissected tissues, which showed different quality of immunofluorescence images, and we chose the best one for TcAK1 (cryosectioning) and TcAK2 (dissected tissues) in our previous manuscript, which resulted in comparatively different images. In the revised manuscript, directly dissected tissues were used for immunofluorescence staining of both TcAK1 and TcAK2, and the results are shown in Figure 4A in revised manuscript.

3.In figure 12, western blot assays should be carried out to test the effects of BR-C on the expression of TcAK1 and TcAK2.

Reply:

According to the reviewer's valuable comment, we performed western blot,

which showed that knockdown of *TcBR-C* up-regulates *TcAK1* expression while down-regulates *TcAK2* expression at both mRNA and protein levels (Figure 5B in revised manuscript).

4. Whether BR-C overexpression in cell line can also influence the expression of AK1 and AK2.

Reply:

We are grateful for this insightful comment. According to reviewer's suggestion, the pAC-DmBR-C-zs was constructed and transfected into the S2 cell line to verify the effects of BR-C on *Drosophila DmAK1* and *DmAK2* expression. We found that overexpression of DmBR-C-Z3 down-regulated *DmAK1* expression, whereas up-regulated *DmAK2*, which is consistent with our findings in *Tribolium castaneum*. However, other BR-C isoforms had no effects on the promoter activities of *DmAKs* (Figure 7 in revised manuscript).

5. In figure 11 and figure 12, the ATP content should to be calculated under the treatment of the hormones and BR-C RNAi.

Reply:

We appreciate this comment. As suggested by the reviewer, we performed the experiments to investigate the effects of 20E and JH III treatments on beetle ATP levels. The results showed knockdown of *TcBR-C* and JH III treatment increased the ATP content, whereas 20E treatment resulted in decrease of ATP content (Supplementary Fig. 11 in supplementary information). The related results of 20E and JH III treatments have been added into the revised manuscript (P12L249-253 in revised manuscript). We also found that knockdown of *TcBR-C* increased the ATP content, however, since *TcBR-C* expression is induced by 20E, and this induction is repressed by JH (Figure 5A, in revised manuscript), we do not think it is necessary to add the BR-C RNAi data into the text.

6. As is known, Kr-h1 transcription factor is the key regulator that mediates the repressive action of JH on insect metamorphosis and inhibits the expression of BR-C, so the authors need to characterize the function of Kr-h1 on *TcAK1* and *TcAK2*.

Reply:

We are grateful for this valuable comment. According to the reviewer's suggestion, RNAi was conducted to investigate the role of *TcKr-h1* in regulation of *TcAK1* and *TcAK2*. The results showed that knockdown of *TcKr-h1* down-regulated the expression of *TcAK1* and up-regulated the expression of *TcAK2* (Figure 6A, P10L201-205 in revised manuscript).

7. Whether several potential Kr-h1 binding sites are predicted on the promoters of *TcAK1* and *TcAK2*?

Reply:

We appreciate this valuable comment. The consensus Kr-h1 binding sites (KBS: TGACCTNNNNYAAC) has been identified by genome-wide ChIP-seq analysis (Kayukawa, T., Jouraku, A., Ito, Y., & Shinoda, T., Molecular mechanism underlying juvenile hormone-mediated repression of precocious larval–adult metamorphosis. Proc Natl Acad Sci U S A, 2017, 114: 1057-1062). However, no putative Kr-h1 binding motifs were found in the promoter regions of *TcAK1* and *TcAK2*.

8. In figure 13, the authors just performed dual-luciferase assay to verify the regulation of BR-C on *TcAK1* and *TcAK2*. Much more methods ought to be used, such as Chip-PCR and EMSA, et al.

Reply:

We are grateful for this insightful comment. As suggested by the reviewer, we have performed EMSA to verify the regulation of BR-C on *TcAK1* and *TcAK2*. The results showed that *TcBR-C-z2* bound directly to the promoters of *TcAK1* and *TcAK2* (Figure 6E, P11L235-238 in revised manuscript).

9. The figures in this manuscript are always piecemeal. It is necessary to renew and integrate the figures.

Reply:

Following the reviewers' suggestion, we have combined related images. Corresponding changes are indicated in the revised manuscript.

10. More recent references regarding to insect BR-C and hormones should be cited.

Reply:

As suggested by the reviewer, we cite several more recent references on insect

BR-C in the revised manuscript (P15L324-325 in revised manuscript), which are highlighted in red in the references section.

Reviewer #2:

1. The developmental expression of both genes during larval development as well as the RNAi analysis conducted on larvae are very difficult to interpret given that larvae are staged by days instead of instar stages. What does 1L, 5L and 22L mean? Given that *Tribolium* larvae undergo 7-8 larval instars and that in each instar there are periods of low and high 20E titers and presence and absence of JH, it is very difficult to interpret the developmental expression levels by merely stating how old are the larvae. Same applies to the RNAi experiments. The authors claim that the dsRNAs are injected in 20-day-old larvae and that mRNA levels for AK1 and AK2 are measured 4 days later. What does 20 days-old mean? Are these larvae in the penultimate larval instar? Are they in the last larval instar? This point is very important to understand the corresponding phenotypes. I suggest to clarify the staging of the samples and different injections throughout the paper by clearly describing in which instar are all the treated animals.

Reply:

We appreciate the reviewer's comment. The staging of the samples has been clarified in Materials and methods section as following: We appreciate the reviewer's comment. The staging of the samples has been clarified in Materials and methods section as following: 1-day-old larvae (1L) corresponds to the early first instar larval stage, 5-day-old larvae (5L) corresponds to the early second instar larval stage, 20-day-old larvae (20L) corresponds to the early last instar larval stage, 22-day-old larvae (22L) corresponds to the mid-late last instar larval stage (P16-17L349-355 in revised manuscript). Additionally, four days after larval RNAi corresponds to the late last instar larval stage.

2. The comparison between AK1 and AK2 mRNA abundance throughout development is based on qPCR, which depends on the efficiency of the primers used in each case. To be sure that the differences are real and can be compared, it is important to check that the efficiency of both primer sets are optimal. If the authors

have analyzed the efficiency of both, AK1 and AK3 primers, with a standard curve then it must be stated in the corresponding section of material and methods section. If not, the authors must run this procedure.

Reply:

We agree with the reviewer's comment. The efficiency of both primer pairs has been evaluated before RT-qPCR, which showed 104.8% for *TcAK1* and 105.0% for *TcAK2*. We have introduced it in the text (P18L373-374 in revised manuscript).

3. The morphological description of larval RNAi phenotype for AK1 is absent. Although clearly affected, it is very difficult to see the morphological defects of AK1-RNAi animals in Fig. 4. A more detailed description of the abnormalities would be necessary. Better and more clear pictures can also help to visualize the phenotype.

Reply:

Following the reviewer's suggestion, we detailed the morphological description of larval RNAi phenotype in the text (P6-7L130-134 in revised manuscript).

4. Regarding the western blot analysis of AK1 and AK2, my main concern is to clarify how these western were obtained. In material and methods, the authors stated that "membranes were incubated with antiAK1 and antiAK2", thus suggesting that both antibodies were incubated simultaneously. If this is the case, and given that both proteins have almost the same molecular weight and the relative position in the membrane of both is almost identical (see Figure S4), it is strange to see that bands for AK1 and AK2 in the western blots (Figs 6, 9 and 11) are very clean and well separated from each other. Given that both bands run at almost the same position, it is very difficult to think that the membranes have been cut before incubating them with the different antibodies. Therefore, to clarify this point, the authors might show the whole membranes for each western blot analysis. On the contrary, if different western blots were carried out for each AK proteins, two different anti-tubulin blots are required for each AK protein western blot.

Reply:

We are sorry that the description of the western blot method caused misunderstanding.

The western blot was conducted separately for TcAK1 and TcAK2 with the same protein samples extracted from different treatment groups. Because the same protein samples were also used for anti-tubulin blots, we think it is not necessary to conduct two different anti-tubulin blots as the internal reference of TcAK1 and TcAK2. Related introduction has been added into the Materials and methods section (P21L451-452 in revised manuscript).

5. Regarding the Br-C-RNAi experiments, some clarifications are required to understand their significance. First, and as stated in point 1, a clear description of the larval stage used is required. This is particularly important in this case as Br-C expression is stage specific with a restricted strong pulse of expression in mid-late last larval instar. Second, expression levels of AK1 and AK2 in dsTcBr-C (without 20E or JH) must be measured as it is important to see how the absence of Br-C affects the expression of both AK genes. Third, and given that Br-C expression is highly affected by the presence of 20E and/or JH as the authors claim in the text, it must be reasoned that the administration of 20E and JH in wt or IB larvae already affects the levels of Br-C in these animals, thus affecting the expression of AK1 and AK2. Therefore, to take this point into consideration, the levels of Br-C must be measured and reported in all treatments presented in Fig 12B. Likewise, the authors can also measure the levels of Kr-h1 as a readout of JH-dependent gene in all cases. Knowing the levels of Br-C and Kr-h1 can clarify the response of AK1 and AK2 to 20E and JH.

Reply:

We are grateful for these insightful comments. As suggested by the reviewer, a clear description of the larval stage used was provided in the Materials and methods section (P16-17L349-355 in revised manuscript).

We also performed RNAi to investigate the effects of knockdown of TcBR-C on the expression levels of *TcAK1* and *TcAK2*. Furthermore, the effects of 20E and JH III on *TcBR-C* expression were also tested. We found that knockdown of *TcBR-C* resulted in the upregulation of *TcAK1* expression, but decreased *TcAK2* expression (A). 20E and JH III treatments showed opposite effects on the expression of TcBR-C. In addition,

knockdown of TcBR-C significantly attenuated the fold changes of *TcAK1* and *TcAK2* expression caused by 20E and JH III treatments (Figure 6B, P11L223-227 in revised manuscript). Considering that the effects of 20E and JH III on *Kr-h1* has been reported in *Tribolium* (Minakuchi C, Namiki T, Shinoda T. Krüppel homolog 1, an early juvenile hormone-response gene downstream of Methoprene-tolerant, mediates its anti-metamorphic action in the red flour beetle *Tribolium castaneum*. Dev Biol. 2009, 32:341-350), we did not measure the levels of *Kr-h1*.

6. Regarding the cell transfection experiments (Fig. 12). Given that Br-C can activate or repress the expression of AK1 and AK2 in the presence/absence of 20E and JH, it would be interesting to mimic the in vivo experiments reported in Fig. 11 in the cell transfection experiments. First, S2 cells transfected with pAC-pGL-TcAK1 or pAC-pGL-TcAK12 must be incubated with 20E and JH alone to measure the expression of AK1 and AK2. Second, S2 cells transfected with pAC-pGL-TcAK1 or pAC-pGL-TcAK12 and PAC5.1-TcBr-C-Z2 (ideally, with all different isoforms) must be incubated with 20E and JH to see if the activation of the promoter behaves as in the in vivo experiments.

Moreover, statistic differences between different PAC5.1-TcBr-C-Z constructs can not be taken into consideration if the levels of each transfected protein are not measured by western blot to normalize the effect by the amount of protein produced.

Reply:

We appreciate these valuable comments. According to the reviewer's suggestion, we transfected s2 cells with the pAC-pGL-TcAK1 or pAC-pGL-TcAK2 reporter vector, followed by treatment of cells with 20E and JH III. The 20E treatment was found to down-regulate the *TcAK1* promoter activity and up-regulate the *TcAK2* promoter activity, while JH III treatment had the opposite effects (Figure 6C, P10L213-216 in revised manuscript). We also transfected S2 cells with pAC-pGL-TcAK1/pAC-pGL-TcAK12 and PAC5.1-TcBr-C-Z2 followed by incubation with 20E and JH, which showed consistency with *in vivo* experiments (Figure 6D, P11L233-237 in revised manuscript).

We also measured the expression of each TcBR-C isoform using an anti-His tag

antibody, and found that the protein expression level of each TcBR-C isoform was similar. Because there are too many figures in this paper, we did not include it in the manuscript, and the data will be provided upon request.

REVIEWERS' COMMENTS:

Reviewer #1 (Remarks to the Author):

The authors have been well addressed my concerns. I think this paper should be accepted for a publication.

Reviewer #2 (Remarks to the Author):

The authors have properly addressed most of the comments and suggestions raised by the reviewer. I believe, therefore, that the work can be considered for publication.